# Clinical Characteristics and Predictors of Long-Term Prognosis of Acute Peripheral Arterial Ischemia Patients Treated Surgically

**DOI:** 10.3390/ijerph20053877

**Published:** 2023-02-22

**Authors:** Piotr Myrcha, Mariusz Kozak, Jakub Myrcha, Mirosław Ząbek, João Rocha-Neves, Jerzy Głowiński, Włodzimierz Hendiger, Witold Woźniak, Izabela Taranta

**Affiliations:** 1Department of General and Vascular Surgery, Faculty of Medicine, Medical University of Warsaw, 02-091 Warsaw, Poland; 2Department of General, Vascular and Oncological Surgery, Masovian Brodnowski Hospital, 03-242 Warsaw, Poland; 3Department of Vascular Surgery and Angiology, Bielanski Hospital, 01-809 Warsaw, Poland; 4Department of Neurosurgery, Centre of Postgraduate Medical Education, 01-826 Warsaw, Poland; 5Department of Angiology and Vascular Surgery, Centro Hospitalar Universitário de São João, 4200-319 Porto, Portugal; 6Department of Biomedicine—Unit of Anatomy, Faculdade de Medicina da Universidade do Porto, 4200-319 Porto, Portugal; 7Department of Vascular Surgery and Transplantalogy, Medical University of Bialystok, 15-276 Bialystok, Poland

**Keywords:** acute peripheral ischemia, atrial fibrillation, mortality, endovascular treatment, thrombectomy, embolectomy

## Abstract

Background: Acute peripheral arterial ischemia is a rapidly developing loss of perfusion, resulting in ischemic clinical manifestations. This study aimed to assess the incidence of cardiovascular mortality in patients with acute peripheral arterial ischemia and either atrial fibrillation (AF) or sinus rhythm (SR). Methods: This observational study involved patients with acute peripheral ischemia treated surgically. Patients were followed-up to assess cardiovascular mortality and its predictors. Results: The study group included 200 patients with acute peripheral arterial ischemia and either AF (n = 67) or SR (n = 133). No cardiovascular mortality differences between the AF and SR groups were observed. AF patients who died of cardiovascular causes had a higher prevalence of peripheral arterial disease (58.3% vs. 31.6%, *p* = 0.048) and hypercholesterolemia (31.2% vs. 5.3%, *p* = 0.028) than those who did not die of such causes. Patients with SR who died of cardiovascular causes more frequently had a GFR <60 mL/min/1.73 m^2^ (47.8% vs. 25.0%, *p* = 0.03) and were older than those with SR who did not die of such causes. The multivariable analysis shows that hyperlipidemia reduced the risk of cardiovascular mortality in patients with AF, whereas in patients with SR, an age of ≥75 years was the predisposing factor for such mortality. Conclusions: Cardiovascular mortality of patients with acute ischemia did not differ between patients with AF and SR. Hyperlipidemia reduced the risk of cardiovascular mortality in patients with AF, whereas in patients with SR, an age of ≥75 years was a predisposing factor for such mortality.

## 1. Introduction

Acute peripheral arterial ischemia is defined as a rapidly developing loss of perfusion, resulting in variable ischemic clinical manifestations and potential necrosis of the involved organ or extremities. This disease is associated with high morbidity and mortality [1]. The incidence of acute limb ischemia is ~1.5 cases per 10,000 persons per year [2]. Diagnostic errors and delays in treatment may lead to the loss of limbs, related to the lack of sufficient time for new blood vessel growth to compensate for the loss of perfusion, or even loss of life [3]. According to previous studies, 15–20% of patients die within the first year after acute lower extremity ischemia, and most of these deaths occur in the peri-operative period. Apart from higher in-hospital mortality, patients with acute arterial ischemia experience adverse events, such as the following: congestive heart failure exacerbation, myocardial infarction, deterioration in renal function, and respiratory complications [4]. There are various causes leading to the occurrence of acute limb ischemia, including the following: arterial embolism (46%), in situ thrombosis (24%), complex factors (20%), and stent- or graft-related thrombosis (10%) [5]. Atrial fibrillation (AF), a sustained cardiac arrhythmia, is the most common cause of embolism and a risk factor for peripheral arterial occlusion [6]. Thromboembolic complications of AF frequently cause morbidity and mortality [7]. Embolism-associated limb ischemia was demonstrated to be related to a higher mortality risk when compared to the occlusion of an artery with local thrombosis in atherosclerotic etiology [8]. Moreover, the mortality risk in patients with AF-related peripheral embolic complications was greater than in those with myocardial infarct-related embolism [9].

The aim of the study is to assess the incidence of cardiovascular mortality in patients with acute peripheral arterial ischemia and either AF or sinus rhythm (SR) and to attempt to identify the predisposing factors of cardiovascular mortality in these two groups of patients during long-term follow-up.

## 2. Materials and Methods

### 2.1. Study Design and Participants

This is a retrospective observational study involving 200 consecutive patients with acute peripheral ischemia and either AF (n = 67) or SR (n = 133), who were admitted to the Department of Vascular Surgery between January 2014 and November 2018. The median follow-up was 21 IQR (7–37) months.

A complete medical history, including information of prior treatment, was obtained from all participants. The diagnosis of acute arterial ischemia was based on the subjective and physical examination and, depending on the area of ischemia, on the different imaging examinations [10]. All patients underwent a duplex ultrasound examination (DUS). Computed tomography angiography (CTA) was performed in 78% of cases of acute limb ischemia and 92% cases of mesenteric ischemia.

AF was diagnosed as defined by the European Society of Cardiology [11]. The risk of thromboembolic complications was assessed with the use of the CHADS_2_ and CHA_2_DS_2_-VASc scales at hospital admission. CHADS2 and CHA_2_DS_2_-VASc scores did not include the thromboembolic event that resulted in hospitalization. The CHADS_2_ score was calculated for each patient in accordance with the following guidelines: congestive heart failure, hypertension, diabetes mellitus, and age ≥75 years were counted as 1 point each; a history of stroke or transient ischemic attack counted as 2 points [12]. The CHA_2_DS_2_-VASc score was also calculated for each patient by current clinical guidelines. This score ranges from 0 to 9 points and includes the following clinical characteristics: congestive heart failure or left ventricular dysfunction (1 point), hypertension (1 point), age ≥75 years (2 points), diabetes mellitus (1 point), prior stroke/transient ischemic attack (TIA) or thromboembolism (2 points each), vascular disease (1 point), age 65–74 years (1 point), and sex category (female; 1 point) [13]. The sum of all factors gives the individual patient’s risk score. In addition, the estimated glomerular filtration rate (eGFR) was calculated using the simplified four-variable Modification of Diet in Renal Disease (MDRD) formula: eGFR = 186 × (serum creatinine)^−1.154^ × (age)^−0.203^ × 0.742 if female] [14].

The study was approved by the university Bioethics Committee (no. 111/2020) and was conducted according to the principles of the Declaration of Helsinki. The university Ethics Committee waived the requirement of obtaining informed consent from the patients.

### 2.2. Surgical Treatment

All patients were treated surgically using open thrombectomy/embolectomy, mechanical thrombectomy (MTH), direct catheter thrombolysis (DCT), or primary amputation. The type of treatment depended on the cause and the depth of ischemia and the patient’s general condition. Revascularisation was not performed in the case of advanced intestinal and limb ischemia/necrosis. In patients with major comorbidities, who experienced significant improvement in their clinical state after conservative treatment, the surgical treatment was postponed for elective preparation. The condition of peripheral circulation was critical in deciding on the treatment method. Due to the lack of peripheral flow in most patients, the implementation of DCT was performed. The restoration of peripheral blood flow was an introduction to other procedures in case of significant stenosis; the treatment was discontinued if proper circulation was restored.

All patients treated with MTH and DCT underwent arteriography. Unfractioned Heparin (UFH) was administered intravenously in all patients with acute ischemia; it was infused with an infusion pump to prolong APTT 2.5–3 times. The infusion was preceded by a bolus of 5000–10,000 IU heparin. In most cases, limb open embolectomy/thrombectomy was performed under epidural or spinal anesthesia. In patients in whom this type of anesthesia was contraindicated, local anesthesia was used along with sedation. The clots were removed using a Fogarty’s catheter, proximally and peripherally, until an acceptable inflow and outflow were obtained. Endarterectomy was performed in the presence of massive atherosclerotic plaques at the site of the artery incision. Arteriotomy was closed with a primary suture or resorting to patch angioplasty in case of a small diameter of the artery. In the absence of a good inflow or outflow, patients were eligible for bypass. In the case of suspected subfascial edema, fasciotomy was performed.

Open visceral thrombectomy/embolectomy was performed by laparotomy, either trans- or retroperitoneal in nature. Patch angioplasty, transposition, or bypass were performed, depending on the etiology of the occlusion. In the case of intestinal necrosis, its resection was performed within the limits of visually healthy tissue. Patients were always qualified for a “second look” within 24–48 h.

### 2.3. Endovascular Treatment

Using WinPepi^®^ V11.65, the required sample for a survival test was computed with a 90% statistical power (β) and a 0.05 significance level [15]. Although bigger event rate disparities are stated, the sample was calculated at 147, with a hazard ratio of 1.6 (1.3 to 1.9) across groups [16,17]. A total estimated sample of 154 was collected with an expected loss-to-follow-up rate of 5%.

DCT and MTH were the first-choice endovascular methods used to treat acute ischemia in all areas [18]. Access via the common femoral or left radial artery was used. Percutaneous transluminal angioplasty (PTA) and stenting were performed. At the time of DCT infusion, Alteplase (Actilyse-Boehringer-Ingelheim^®^) 1 mg/h was administered (5 mg bolus). UFH was administered simultaneously to the sheath (500 IU/h). The fibrinogen and APTT levels were set four times a day. DCT was terminated earlier if the fibrinogen level fell below 150 mg/dL [19,20]. Control arteriography was performed before sheath removal. During mechanical thrombectomy, AngioJet (Boston Scientific, Marlborough, MA, USA) and Rotarex (Straub Medical, Vilters-Wangs, Switzerland) systems were used with different catheter diameters, depending on the size of the artery. If the procedure’s effectiveness was insufficient, DCT or PTA/stent was performed. The Spider embolic protection system (Medtronic) was used during some procedures on the arteries of the lower limbs. After the surgery, UFH was administered intravenously using an infusion pump with the target of prolonging APTT 2.5–3 times. In the case of simultaneous occlusion of the celiac trunk and superior mesenteric artery, we tried to open both arteries. DCT was used carefully because of the known mechanism of endogenous thrombolysis occurring during intestinal ischemia [21].

### 2.4. Study Endpoint

The study endpoint was cardiovascular mortality during long-term observation.

### 2.5. Statistical Analysis

Categorical data are expressed as numbers of patients and percentages. The Chi-squared test or Fisher’s exact test were used to compare proportions. Numeric variables are presented as medians and quartiles and compared using the Mann–Whitney U test, because their distribution was not normal (assessed by the Shapiro–Wilk test, graphical curve analysis and kurtosis). In the context of survival analysis, the endpoint was defined as cardiovascular death. The follow-up was calculated as the number of days from surgery to death (cardiovascular or not) or to the end of the study (for live patients). Survival curves for AF and SR groups were created by the Kaplan–Meier method.

Patients’ characteristics and type of surgery were assessed in univariable Cox proportional hazards regression models to evaluate the relationship with cardiovascular death. The regressive predictive model was created by resorting to regression analysis and dimension reduction by the method of backward feature elimination. Variables with clinical relevance included in the multivariate analysis were associated with the group including cardiovascular and non-cardiovascular death in the univariate analysis, with statistical significance *p* < 0.1. Some multivariable Cox proportional hazards models are also presented. Hazard ratios (HR) in univariable and multivariable Cox models were estimated, along with 95% confidence intervals. A stratified analysis was conducted for SR and AF patients. Cox proportional hazards regression models were not created for categorical variables with less than five patients in any category. Statistical tests were two-tailed, and *p*-values < 0.05 were considered significant. All statistical analyses were performed using the R software package version 3.6.2.

## 3. Results

### 3.1. Characteristics of the Study Group

In the present study of 200 patients, 67 (33.5%) had AF and 133 (66.5%) had SR. Patients with AF were statistically significantly older (78.0 vs. 70.0, *p* =0.003) and women represented the majority (62.7% vs. 39.1%, *p* = 0.002) in this group, in comparison to the group with SR. The incidence of comorbidities did not differ significantly between groups; only ischemic heart disease was more prevalent in the group with AF (55.2% vs. 35.3%, *p* = 0.007). Patients with AF had statistically lower eGFR than patients with SR (64.3 vs. 73.7, *p* = 0.03). In this group of patients, the results of CHADS_2_ and CHA_2_DS_2_-VASc were also higher than in patients with SR. In patients with AF, embolic occlusion was more frequent, while in patients with SR, the occlusion was associated with thrombotic material. Thromboembolic material was observed in both groups, predominantly in the lower limbs (74.6% vs. 84.2%, *p* = 0.10). The clinical characteristics of patients with AF and SR are presented in Table 1.

### 3.2. Incidence of Mortality in Patients with Acute Peripheral Arterial Ischemia and Atrial Fibrillation or Sinus Rhythm

The median follow-up in the group with AF was 20.9 (IQR: 7.4, 34.3) months, and in the group with SR it was 22.6 (IQR: 7.4, 40.3) months, *p* = 0.45. There were no differences in all-cause mortality between the AF group and SR group (43.3% vs. 31.6%, *p* = 0.10). Cardiovascular mortality was similar in patients with AF and SR (28.4% vs. 18.8%, *p* = 0.12) (Table 2).

The analysis of Kaplan–Meier curves shows that in the initial period after surgery, the chances of survival were similar in both groups (Figure 1).

### 3.3. Factors Predisposing to Cardiovascular Mortality

In the group with AF, in patients who died of cardiovascular causes, the prevalence of PAD and hypercholesterolemia was lower than in those who did not die of such causes (PAD 31.6% vs. 58.3%, *p* = 0.048; hypercholesterolemia 5.3% vs. 31.2%, *p* = 0.03) (Table 3).

The comparison of patients with SR who died of cardiovascular causes and those with SR who did not die of such causes revealed that the first group more frequently had GFR < 60 mL/min/1.73 m^2^ (47.8% vs. 25.0%, *p* = 0.03), and they tended to be older (age >75 years: 60.0% vs. 33.3%, *p* = 0.04) (Table 4).

In this study, the CHA_2_DS_2_-VASc score was similar in patients who died of cardiovascular causes and in those who did not die of such causes, both in the AF and in the SR group.

The multivariable analysis showed that the presence of hyperlipidemia reduced the risk of cardiovascular mortality in patients with AF, whereas in the case of patients with SR, an age of ≥75 years was the factor predisposing one for such mortality (Table 5).

## 4. Discussion

AF increases the risk of thromboembolic episodes, which are often responsible for high morbidity and mortality in this group of patients [7,22,23]. The study of Barreto et al. [24], comprising patients with peripheral arterial embolism, confirmed the role of AF in the pathogenesis of acute limb ischemia. In our study, the estimated glomerular filtration rate (eGFR) of AF patients was significantly lower than in SR patients. Moreover, they had higher CHA_2_DS_2_-VASc scoring in comparison to patients with sinus rhythms. Despite this, only 35.8% of patients in the AF group were receiving oral anticoagulants, and even fewer were treated with antiplatelet agents (13.4%) before hospital admission. In turn, SR patients were significantly more often administered antiplatelet treatment (APT) (40.6%), which is in accordance with current recommendations. Howard et al. [5] demonstrated that premorbid levels of anticoagulation in patients suffering from acute events of cardioembolic origin, as well as known AF, are deficient. However, the vast majority of patients with a high thromboembolism risk (CHA_2_DS_2_-VASc scores ≥ 2) had no contraindications to anticoagulation. Additionally, Ralevic et al. [25], in a prospective observational study of consecutive patients with lower limb amputation, found that despite a high prevalence of AF, patients often did not receive the recommended oral anticoagulation therapy. In this study, the occlusion in patients with AF and SR was mainly localized to the lower extremities. The higher prevalence of acute lower limb ischemia has also been indicated in other studies. Ischemia affecting upper extremities is relatively uncommon, accounting for less than 5% of all cases of limb ischemia [26,27]. In this study, acute peripheral arterial ischemia in AF patients was primarily caused by an embolus (65.7%), and in SR patients by a thrombus (55.6%). This observation was confirmed by Mutirangura et al. [28], who revealed that AF was more prevalent in patients with acute arterial embolism than acute arterial thrombosis.

Systematic reviews and meta-analyses have undoubtedly indicated the association of AF with an increased risk of mortality in patients with coronary artery disease [29,30]. However, the prognostic implication of AF in acute peripheral arterial ischemia has not been extensively studied. We did not observe statistically significant differences in mortality between patients with AF and patients with SR, who were operated on due to acute ischemia. Cardiovascular mortality was slightly higher in patients with AF compared to those with SR (28.4% vs. 18.8%); however, this difference failed to reach statistical significance. A similar trend was observed in the study of Ralevic et al. al. [25], who demonstrated that lower limb amputation, cardiovascular death, as well as adverse cardiovascular events were more common in patients with AF during follow-up compared with patients without AF. Lorentzen et al. [31] showed that AF enhanced the risk of mortality, decreased patients’ quality of life, and increased the number of hospitalizations. Moreover, according to Vohra et al. [32], in patients with AF-related peripheral embolic complications, the mortality risk was higher compared to individuals with embolism associated with myocardial infarction. Data from the Reduction of Atherothrombosis for Continued Health Registry [33] demonstrated that AF was an independent predictor of long-term CV events in patients with symptomatic peripheral arterial disease (PAD) [34]. We can only suspect that the lack of statistically significant differences in mortality between AF and SR patients in our study is associated with the relatively small number of AF patients, as well as with the introduction of appropriate treatment, because, as mentioned above, before the hospitalization, many patients were not receiving the best medical treatment.

In this study, we also observed that the CHA_2_DS_2_-VASc scale was not a predictor of cardiovascular mortality in patients with AF and SR. In general, the CHA_2_DS_2_-VASc score can be used to assess the risk of stroke in patients with atrial fibrillation. However, published results show considerable variability in relation to the mortality of AF patients and the correlation with the CHA_2_DS_2_-VASc score, especially regarding patient history, drug treatment, and clinical status [35,36,37]. In an observational retrospective cohort study (CONSORT compliant), the predictive value of CHA_2_DS_2_-VASc was confirmed in relation to overall all-cause mortality [36]. Patients with higher risk scores had a survival rate of 79.1%, while medium-risk and low-risk patients had survival rates of 95.6% and 100%, respectively. According to Potpara et al. [38], the CHA_2_DS_2_-VASc score is a reliable predictor of 30-day unfavorable outcomes of patients with acute ischemic stroke. Its sensitivity and specificity for unfavorable short-term functional outcomes is greater in comparison to other scores, including the CHADS_2_ and HAS-BLED (93.5% vs. 92.4% vs. 71.7% and 77.0% vs. 61.5% vs. 69.6%, respectively; all *p* < 0.05). Despite the CHA_2_DS_2_-VASc score differing significantly between the AF and SR groups in our study, it did not correlate with cardiovascular mortality, probably due to the fact that many more patients with AF received appropriate treatment before the hospitalization, which might have influenced their outcomes. The presence of AF may be a primary driver of the administration of therapy for stroke prevention, which decreases mortality rate. Jackson et al. [35] confirmed that systemic oral anticoagulant treatment (OAC) was associated with lower rates of all-cause mortality, cardiovascular death, and first stroke/TIA among patients with CHA_2_DS_2_-VASc score ≥ 2. Also, in most of studies, this scale was used to assess all-cause death, not cardiovascular mortality.

A univariable analysis of factors modulating the risk of cardiovascular mortality in our population of patients with acute peripheral arterial ischemia and AF demonstrated that PAD and hypercholesterolemia (obesity paradox) reduced the risk of cardiovascular mortality. According to numerous studies, PAD and AF share similar epidemiologic patterns and risk factors, and their presence is related to increased morbidity and mortality [39]. A sub-analysis performed with the use of data from the Reduction of Atherothrombosis for Continued Health Registry demonstrated that the combined presence of AF and PAD significantly increased the rates of cardiovascular (CV) death [33]. Additionally, Lin et al. [40] found that the coexistence of AF and PAD considerably enhanced the risk for all major adverse outcomes, and it was associated with at least a two-fold higher risk of CV death than in patients with AF or PAD only. The reduction in cardiovascular mortality related to the presence of PAD in AF patients in this study may be associated with the fact that PAD patients were probably previously intensively treated with antihypertensive, lipid-lowering, and antiplatelet drugs. Indeed, 94.7% of patients in this group used antiplatelet drugs. It is also possible that patients with an earlier diagnosis of PAD introduced dietary changes and ceased smoking, thus decreasing their cardiovascular risk. The importance of hypercholesterolemia as a factor in reducing cardiovascular mortality in the group with AF was also confirmed in a multivariable analysis, in which it decreased the risk of death by 87%. Again, such a phenomenon could be associated with the fact that patients with a history of hyperlipidemia were treated with statins and other lipid lowering drugs, which reduced their cardiovascular mortality. Additionally, Clua-Espuny et al. [41] found that mortality among AF patients was significantly lower for those treated with statins. The obesity paradox in atrial fibrillation patients, particularly for all-cause and cardiovascular death outcomes, has been extensively described [42,43].

In the case of patients with SR, the univariable analysis revealed a correlation between cardiovascular mortality and age. The risk of cardiovascular mortality increased 1.6 times with every ten years. Additionally, in a retrospective review of patients with acute limb ischemia, the risk of mortality increased with age and renal failure, but also with the female gender, cancer, in situ thrombosis or embolic etiology, cardiac events, and hemorrhagic events [44]. Eliason et al. [45] indicated that in patients with acute lower extremity ischemia, an age of less than 63 years was an independent variable associated with a decreased risk of in-hospital mortality. Finally, the analysis of the National Audit of Thrombolysis for Acute Leg Ischemia (NATALI) database confirmed that the mortality of patients who had undergone intra-arterial thrombolysis to treat acute leg ischemia was higher in women and older patients, and in patients with native vessel occlusion, emboli, or a history of ischemic heart disease [46]. The relationship between higher mortality and advanced age may be due to the fact that the prevalence of comorbidities increases with advancing age in many populations. The impact of age was also confirmed in our multivariate analysis.

We also observed that in patients with SR and eGFR < 60 mL/min, the risk of cardiovascular death was 2.48-fold higher when compared to those with higher eGFR. In turn, Kuoppala et al. [47] demonstrated that renal insufficiency was among the independent factors associated with in-hospital mortality after thrombolysis. Moreover, they indicated that, among other reasons, renal insufficiency and an age ≥80 years were associated with mortality during follow-up. They suggested that the administration of a contrast agent during angiography may be partly responsible for such a negative relationship in this subgroup of patients. Renal impairment is also more frequent and aggravated in patients with CAD and vascular complications, which can also explain the association between lower GFR and cardiovascular mortality. Maithel et al. [48] confirmed the relationship between renal insufficiency and poorer outcomes in patients after open vascular surgery. Additionally, in patients with AF, renal dysfunction proved to be a strong, independent predictor of left atrial appendage thrombus formation [49]. This study has demonstrated that acute peripheral arterial ischemia continues to be associated with high mortality despite advances in endovascular-based therapies and improved critical care.

The choice of the treatment method in patients with acute limb ischemia is difficult. There are no strict criteria defining the risk of reperfusion syndrome after revascularization. Studies on the preoperative inflammatory biomarkers’ neutrophil-to-lymphocyte ratio and platelet-to-lymphocyte ratio are encouraging. Increased preoperative values of these factors may be indicators of a poor outcome and the need for primary amputation [50].

A significant limitation of the study is the small size of the study group. Patients were treated and followed in a single tertiary care center with a high volume, which might affect the external validity of the results. Other limitations involve the lack of data on non-anticoagulant/antiplatelet and other therapy before hospitalization, the lack of detailed information on the post-surgery period, and the lack of data on ischemic events during the follow-up period.

## 5. Conclusions

The mortality of patients operated on due to acute ischemia did not differ significantly between the group of patients with AF and those with SR. Moreover, the CHA_2_DS_2_-VASc scale proved not to be a good predictor of cardiovascular mortality in patients with AF and SR. The presence of hyperlipidemia reduced the risk of cardiovascular mortality in patients with AF, whereas in the case of patients with SR, an age of ≥75 years was a factor predisposing one to such mortality.

## Figures and Tables

**Figure 1 ijerph-20-03877-f001:**
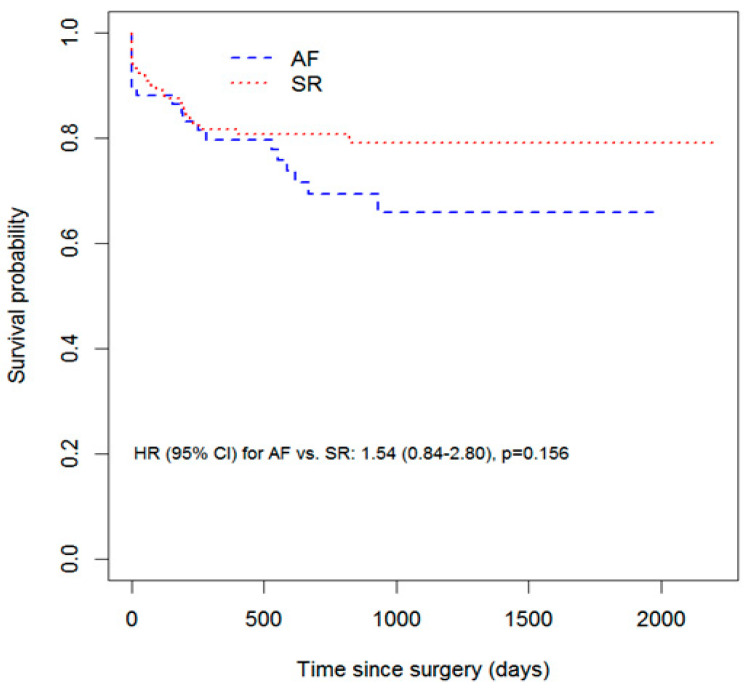
Kaplan–Meier curve for cardiovascular mortality in patients with atrial fibrillation and sinus rhythm.

**Table 1 ijerph-20-03877-t001:** Characteristics of study groups.

Clinical Characteristics	Atrial Fibrillation n = 67	Sinus Rhythm n = 133	*p*-Value
Female	42 (62.7)	52 (39.1)	**0.002**
Age, years, median (Q1, Q3)	78.0 (68.5, 82.0)	70.0 (62.0, 80.0)	**0.003**
Medical history			
Coronary artery disease	37 (55.2)	47 (35.3)	**0.007**
Myocardial infarction	12 (17.9)	23 (17.3)	0.91
PCI	6 (9.0)	13 (9.8)	0.85
CABG	3 (4.5)	4 (3.0)	0.69
Vascular disease	39 (58.2)	92 (69.2)	0.12
PAD	34 (50.7)	83 (62.4)	0.11
Carotid stenosis	1 (1.5)	6 (4.5)	0.43
Hypertension	54 (80.6)	91 (68.4)	0.07
Heart failure	24 (35.8)	32 (24.1)	0.08
Hyperlipidemia	16 (23.9)	25 (18.8)	0.40
Diabetes mellitus	16 (23.9)	34 (25.6)	0.80
COPD	3 (4.5)	7 (5.3)	1
TIA	0 (0.0)	3 (2.3)	0.55
Stroke ischemic	14 (20.9)	18 (13.5)	0.18
Stroke hemorrhagic	3 (4.5)	4 (3.0)	0.69
Deep vein thrombosis	2 (3.0)	0 (0.0)	0.11
Pulmonary embolism	1 (1.5)	1 (0.8)	1
Alcohol abuse	4 (6.0)	8 (6.0)	1
Nicotine	16 (23.9)	42 (31.6)	0.26
CHADS_2_-median (Q1, Q3)	2.0 (1.0, 4.0)	2.0 (1.0, 2.0)	**0.003**
CHA_2_DS_2_-VASc -median (Q1, Q3)	4.0 (3.0, 6.0)	3.0 (2.0, 5.0)	**<0.001**
GFR, ml/min-median (Q1, Q3)	64.3 (45.0, 79.1)	73.7 (56.2, 97.8)	**0.03**
GFR < 60 mL/min	28 (48.3)	33 (29.7)	**0.02**
Anticoagulant treatment			
OAC	24 (35.8)	1 (0.8)	**<0.001**
APT	9 (13.4)	54 (40.6)	**<0.001**
No treatment	30 (44.8)	74 (55.6)	0.14
Location of thromboembolic material			
Upper extremity	15 (22.4)	14 (10.5)	**0.02**
Lower extremity	50 (74.6)	112 (84.2)	**0.10**
Acute mesenteric ischemia	2 (3.0)	7 (5.3)	0.72
The cause of ischemia			
Embolus	44 (65.7)	59 (44.4)	**0.004**
Thrombus	23 (34.3)	74 (55.6)	**0.004**
Type of surgery			
Open thrombectomy	55 (82.1)	96 (72.2)	0.12
MTH	9 (13.4)	14 (10.5)	0.54
DCT	3 (4.5)	21 (15.8)	**0.02**
Amputation	0 (0.0)	2 (1.5)	0.55

APT—antiplatelet treatment, CABG—coronary artery bypass grafting, COPD—chronic obstructive pulmonary disease, DCT—direct catheter thrombolysis, GFR—glomerular filtration rate, MTH—mechanical thrombectomy, CHADS_2_- score for fibrilation stroke risk, CHA_2_DS_2_-VASc- modified score for fibrilation stroke risk OAC—oral anticoagulant, PAD—peripheral artery disease, PCI—percutaneous coronary intervention, TIA—transient ischemic attack. *p*-Values in bold indicate statistical significance

**Table 2 ijerph-20-03877-t002:** Comparison of mortality of patients with atrial fibrillation and sinus rhythm.

	Atrial Fibrillation n = 67	Sinus Rhythm n = 133	*p*-Value
All-cause mortality	29 (43.3)	42 (31.6)	0.10
Mortality during hospitalization	10 (14.9)	15 (11.3)	0.46
Mortality after hospitalization	19 (28.4)	27 (20.3)	0.20
Mortality within a year of surgery	20 (29.9)	38 (28.6)	0.85
Mortality from cardiovascular causes	19 (28.4)	25 (18.8)	0.12

**Table 3 ijerph-20-03877-t003:** The comparison of patients with atrial fibrillation who died of cardiovascular causes with those who did not die of such causes.

Clinical Characteristics	All Patients n = 67	Cardiovascular Deaths n = 19	No Cardiovascular Deaths n = 48	*p*-Value
Female	42 (62.7)	14 (73.7)	28 (58.3)	0.24
Age, years, median (Q1–Q3)	78.0 (68.5, 82.0)	77.0 (64.0, 81.0)	78.0 (71.8, 83.2)	0.44
Medical history				
Coronary artery disease	37 (55.2)	9 (47.4)	28 (58.3)	0.41
Myocardial infarction	12 (17.9)	3 (15.8)	9 (18.8)	1
PCI	6 (9.0)	2 (10.5)	4 (8.3)	1
CABG	3 (4.5)	1 (5.3)	2 (4.2)	1
Vascular disease	39 (58.2)	9 (47.4)	30 (62.5)	0.26
PAD	34 (50.7)	6 (31.6)	28 (58.3)	**0.048**
Carotid stenosis	1 (1.5)	1 (5.3)	0 (0.0)	0.28
Hypertension	54 (80.6)	14 (73.7)	40 (83.3)	0.49
Heart failure	24 (35.8)	8 (42.1)	16 (33.3)	0.50
Hyperlipidemia	16 (23.9)	1 (5.3)	15 (31.2)	**0.03**
Diabetes mellitus	16 (23.9)	5 (26.3)	11 (22.9)	0.76
COPD	3 (4.5)	2 (10.5)	1 (2.1)	0.19
Stroke ischemic	14 (20.9)	4 (21.1)	10 (20.8)	1
Stroke hemorrhagic	3 (4.5)	1 (5.3)	2 (4.2)	1
Deep vein thrombosis	2 (3.0)	0 (0.0)	2 (4.2)	1
Pulmonary embolism	1 (1.5)	0 (0.0)	1 (2.1)	1
GFR, mL/min-median (Q1, Q3)	64.3 (45.0, 79.1)	55.7 (46.8, 70.3)	67.6 (45.7, 80.4)	0.57
GFR < 60 mL/min	28 (48.3)	8 (57.1)	20 (45.5)	0.45
CHADS_2_-median (Q1, Q3)	2.0 (1.0, 4.0)	2.0 (1.5, 4.0)	2.0 (1.0, 4.0)	0.88
CHA_2_DS_2_-VASc-median (Q1, Q3)	4.0 (3.0, 6.0)	5.0 (3.0, 6.0)	4.0 (3.8, 5.2)	0.98
Anticoagulant treatment				
OAC	24 (35.8)	5 (26.3)	19 (39.6)	0.30
APT	9 (13.4)	1 (5.3)	8 (16.7)	0.43
No treatment	30 (44.8)	11 (57.9)	19 (39.6)	0.17
Location of thromboembolic material				
Upper extremity	15 (22.4)	5 (26.3)	10 (20.8)	0.75
Lower extremity	50 (74.6)	14 (73.7)	36 (75.0)	1
Acute mesenteric ischemia	2 (3.0)	0 (0.0)	2 (4.2)	1
The cause of ischemia				
Embolus	44 (65.7)	15 (78.9)	29 (60.4)	0.15
Thrombus	23 (34.3)	4 (21.1)	19 (39.6)	0.15
Type of surgery				
Open thrombectomy	55 (82.1)	18 (94.7)	37 (77.1)	0.16
MTH	9 (13.4)	1 (5.3)	8 (16.7)	0.43
DCT	3 (4.5)	0 (0.0)	3 (6.2)	0.55

APT—antiplatelet treatment, CABG—coronary artery bypass grafting, COPD—chronic obstructive pulmonary disease, DCT—direct catheter thrombolysis, GFR—glomerular filtration rate, MTH—mechanical thrombectomy, CHADS_2_- score for fibrilation stroke risk, CHA_2_DS_2_-VASc- modified score for fibrilation stroke risk, OAC—oral anticoagulant, PAD—peripheral artery disease, PCI—percutaneous coronary intervention, TIA—transient ischemic attack. *p*-Values in bold indicate statistical significance

**Table 4 ijerph-20-03877-t004:** The comparison of patients with sinus rhythm who died of cardiovascular causes with those who did not die of such causes.

Clinical Characteristics	All Patients n = 133	Cardiovascular Deaths n = 25	No Cardiovascular Deaths n = 108	*p*-Value
Female	52 (39.1)	13 (52.0)	39 (36.1)	0.14
Age, years, median (Q1, Q3)	70.0 (62.0, 80.0)	79.0 (65.0, 86.0)	68.5 (60.8, 78.2)	**0.02**
Medical history				
Coronary artery disease	47 (35.3)	9 (36.0)	38 (35.2)	0.94
Myocardial infarction	23 (17.3)	6 (24.0)	17 (15.7)	0.38
PCI	13 (9.8)	3 (12.0)	10 (9.3)	0.71
CABG	4 (3.0)	1 (4.0)	3 (2.8)	0.57
Vascular disease	92 (69.2)	16 (64.0)	76 (70.4)	0.53
PAD	83 (62.4)	15 (60.0)	68 (63.0)	0.78
Carotid stenosis	6 (4.5)	2 (8.0)	4 (3.7)	0.31
Hypertension	91 (68.4)	17 (68.0)	74 (68.5)	0.96
Heart failure	32 (24.1)	8 (32.0)	24 (22.2)	0.30
Hyperlipidemia	25 (18.8)	2 (8.0)	23 (21.3)	0.16
Diabetes mellitus	34 (25.6)	7 (28.0)	27 (25.0)	0.76
COPD	7 (5.3)	1 (4.0)	6 (5.6)	1
Stroke ischemic	18 (13.5)	3 (12.0)	15 (13.9)	1
Stroke hemorrhagic	4 (3.0)	1 (4.0)	3 (2.8)	0.57
Pulmonary embolism	1 (0.8)	0 (0.0)	1 (0.9)	1
GFR, ml/min-median (Q1, Q3)	73.7 (56.2, 97.8)	61.3 (27.9, 90.5)	76.5 (59.4, 100.2)	0.06
GFR < 60 mL/min	33 (29.7)	11 (47.8)	22 (25.0)	**0.03**
CHADS_2_-median (Q1, Q3)	2.0 (1.0, 2.0)	2.0 (1.0, 3.0)	2.0 (1.0, 2.0)	0.30
CHA_2_DS_2_-VASc-median (Q1, Q3)	3.0 (2.0, 5.0)	4.0 (3.0, 5.0)	3.0 (2.0, 5.0)	0.13
Anticoagulant treatment				
OAC	1 (0.8)	1 (4.0)	0 (0.0)	0.19
APT	54 (40.6)	11 (44.0)	43 (39.8)	0.70
No treatment	74 (55.6)	13 (52.0)	61 (56.5)	0.68
Location of thromboembolic material				
Upper extremity	14 (10.5)	2 (8.0)	12 (11.1)	1
Lower extremity	112 (84.2)	22 (88.0)	90 (83.3)	0.76
Acute mesenteric ischemia	7 (5.3)	1 (4.0)	6 (5.6)	1
The cause of ischemia				
Embolus	59 (44.4)	11 (44.0)	48 (44.4)	0.97
Thrombus	74 (55.6)	14 (56.0)	60 (55.6)	0.97
Type of surgery				
Open thrombectomy	96 (72.2)	22 (88.0)	74 (68.5)	0.05
MTH	14 (10.5)	3 (12.0)	11 (10.2)	0.73
DCT	21 (15.8)	0 (0.0)	21 (19.4)	**0.01**

APT—antiplatelet treatment, CABG—coronary artery bypass grafting, COPD—chronic obstructive pulmonary disease, DCT—direct catheter thrombolysis, GFR—glomerular filtration rate, MTH—mechanical thrombectomy, CHADS_2_- score for fibrilation stroke risk, CHA_2_DS_2_-VASc- modified score for fibrilation stroke risk, OAC—oral anticoagulant, PAD—peripheral artery disease, PCI—percutaneous coronary intervention, TIA—transient ischemic attack. *p*-Values in bold indicate statistical significance

**Table 5 ijerph-20-03877-t005:** Univariable and multivariable logistic regression analysis—factors predisposing one to cardiovascular mortality in the group of patients with atrial fibrillation and sinus rhythm.

Patients with Atrial Fibrillation						
	Univariable Analysis			Multivariable Analysis		
	HR	95%CI	*p*-Value	aHR	95%CI	*p*-Value
Hyperlipidemia	0.43	0.16–1.12	0.08	0.13	0.02–1.00	**0.049**
APT	0.31	0.04–2.30	0.25	0.28	0.04–2.10	0.22
Patients with Sinus Rhythm						
	Univariable Analysis			Multivariable Analysis		
	HR	95%CI	*p*-Value	aHR	95%CI	*p*-Value
Age 75 or more years	3.45	1.25–9.51	**0.02**	2.75	1.09–7.00	**0.03**
GFR (per 10 units)	0.90	0.80–1.02	0.09	0.96	0.84–1.09	0.53

APT—antiplatelet treatment, CI—confidence interval, GFR—glomerular filtration rate, aHR—adjusted hazard ratio. *p*-Values in bold indicate statistical significance.

## Data Availability

The data that support the findings of this study are available from the corresponding author upon reasonable request.

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
