# Peer review of "Clinical Characteristics and Predictors of Long-Term Prognosis of Acute Peripheral Arterial Ischemia Patients Treated Surgically"

_ijerph, 2023, doi:10.3390/ijerph20053877_

Round 1
Reviewer 1 Report
Major revision is required.
1. Methods:
(1) Was this study a retrospective or prospective design? Please clarify it.
(2) Please clearly describe the end of follow-up time in this study.
(3) Most importantly, the minimum sample size of this study should be calculated. If this study cannot achieve the minimum sample size, you cannot make any conclusions.
(4) The drawback of this study was statistical analysis. The distribution of this study was not normal, thereby Cox proportional hazard regression assumption was not met and Cox proportional hazard regression should not be used. In this setting, you should use restricted mean survival time for survival analysis and subgroup analysis.
2. Results:
(1) Page 6, line 205: “higher” should be corrected to “lower”.
(2) In the table 5, please list adjusted covariates in the footnote.
Author Response
- Methods:
(1) Was this study a retrospective or prospective design? Please clarify it.
This study presents a retrospective design. Adjustments were added to the manuscript
Please see page 2 line 71
“This is a retropective observational study involving 200 consecutive patients with acute peripheral ischemia and either AF (n=67) or SR (n=133), who were admitted to the Department of Vascular Surgery between January 2014 and November 2018.”
(2) Please clearly describe the end of follow-up time in this study.
The authors agree with the reviewer. The median follow-up was added to the manuscript.
Please see page 2 line 73
The median follow-up was 21 IQR [7-37] months
(3) Most importantly, the minimum sample size of this study should be calculated. If this study cannot achieve the minimum sample size, you cannot make any conclusions.
The authors agree with the reviewer. A sample size estimation would add value and qualify this study for high quality metanalysis.
Please see page 3 line 124-127
Using WinPepi® V11.65, the required sample for a survival test was computed with a 90% statistical power (β) and a 0.05 significance level. Although bigger event rate disparities are stated, the sample was calculated at 147, with a hazard ratio of 1.6 (1.3 to 1.9) across groups. A total estimated sample of 154 was collected with an expected loss-to-follow-up rate of 5%.
(4) The drawback of this study was statistical analysis. The distribution of this study was not normal, thereby Cox proportional hazard regression assumption was not met and Cox proportional hazard regression should not be used. In this setting, you should use restricted mean survival time for survival analysis and subgroup analysis.
The authors used the statistical assumption, that since the sample is over 50 patients, normality can be assumed, since no major skewness in distribution was observed. It is also usual in previous evidence that CHadsvasc does not have a perfectly normal distribution since the ones with the higher score are more prone to death.
An exception is made for survival analysis, since due to issues of recruitment it is very rare to have a normal distribution, thus being reported in the general literature has median survival and not mean survival. The authors believe that the assumptions are fulfilled, and a strong test as a cox hazard multivariate analysis can provide significant a valuable result.
- Results:
(1) Page 6, line 205: “higher” should be corrected to “lower”.
Done
(2) In the table 5, please list adjusted covariates in the footnote.
The following amendtment was added to the methods
Please see page 4 line 154-157
Variables with clinical relevance included in the multivariate analysis were associated with the group with cardiovascular and non-cardiovascular death in the univariate analysis, with statistical significance P < 0.1.

Reviewer 2 Report
I would like to congratulate the authors for their work. This is potentially significant research because it provides evidence of cardiovascular mortality in ALI patients between AF and SR patients. The authors' conclusions were that hyperlipidemia reduced the risk of mortality in patients with AF, and age over 75 years was a predictor of mortality in patients with SR.
As we see in Table 1, female, age, coronary artery disease, CHADS2, CHA2DS2-VASc, location of thromboembolic material and cause of ischemia have a higher incidence in patients with AF. I suggest the authors modify the univariable and multivariable analysis and enter all factors with a statically significant difference between the 2 groups, for all enrolled patients.
Furthermore, it is well known that patients with acute limb ischemia have a higher risk of amputation, it will be very interesting if the authors can enter the risk of amputation and subsequently enter the new data into the multivariate analysis.
Authors can improve the quality of their manuscript by comparing the output with the following recently published article in the literature:
- https://doi.org/10.3390/life12060822
- https://doi.org/10.3390/jcm11082116
- https://doi.org/10.3390/jcm11071936
Author Response
I would like to congratulate the authors for their work. This is potentially significant research because it provides evidence of cardiovascular mortality in ALI patients between AF and SR patients. The authors' conclusions were that hyperlipidemia reduced the risk of mortality in patients with AF, and age over 75 years was a predictor of mortality in patients with SR.
We would like to express the sincerest gratitude for taking the time to review our manuscript
As we see in Table 1, female, age, coronary artery disease, CHADS2, CHA2DS2-VASc, location of thromboembolic material and cause of ischemia have a higher incidence in patients with AF. I suggest the authors modify the univariable and multivariable analysis and enter all factors with a statically significant difference between the 2 groups, for all enrolled patients.
The authors agree and understand the doubts exposed by the reviewer. Further details on how the analysis was performed were added.
Adjustments were added to the manuscript – please see line 155-159
“The regressive predictive model was created resorting to regression analysis and dimension reduction by the method of backward feature elimination. Variables with clinical relevance included in the multivariate analysis were associated with the group with cardiovascular and non-cardiovascular death in the univariate analysis, with statistical significance P < 0.1. “
Furthermore, it is well known that patients with acute limb ischemia have a higher risk of amputation, it will be very interesting if the authors can enter the risk of amputation and subsequently enter the new data into the multivariate analysis.
Authors can improve the quality of their manuscript by comparing the output with the following recently published article in the literature:
- https://doi.org/10.3390/life12060822
- https://doi.org/10.3390/jcm11082116
- https://doi.org/10.3390/jcm11071936
In response to the request, we have carefully reviewed our bibliography and have made the necessary additions to include the articles you suggested. These additional references will provide a more comprehensive and well-rounded perspective on the subject matter.

Reviewer 3 Report
The manuscript is original and of scientific interest for the involved clinicians, taking into account the information in this field and the increasing incidence of atrial fibrillation and cardiovascular risk factors as causes of embolic and thrombotic events, respectively. The study also highlights the importance of the inappropriate anticoagulant treatment, generating vascular complications and high mortality.
The current study has a clear objective and design, the methodology is well described, the results are clearly illustrated by tables and figures. I especially appreciate the section of discussion which is well organized and it is sustained by updated data as reflected in the selected references.
I think it is commendable that the authors have chosen this theme for clinical research, as acute peripheral arterial ischemia is a pathology often underdiagnosed and undertreated adequately with potential fatal cardiovascular consequences.
I agree that the small size of population studied it is a significant limitation of the study, but I encourage the authors to extend the group of this category of patients, may be in a collaborative study and to consider this research as a pilot study.
Author Response
The manuscript is original and of scientific interest for the involved clinicians, taking into account the information in this field and the increasing incidence of atrial fibrillation and cardiovascular risk factors as causes of embolic and thrombotic events, respectively. The study also highlights the importance of the inappropriate anticoagulant treatment, generating vascular complications and high mortality.
The current study has a clear objective and design, the methodology is well described, the results are clearly illustrated by tables and figures. I especially appreciate the section of discussion which is well organized, and it is sustained by updated data as reflected in the selected references.
I think it is commendable that the authors have chosen this theme for clinical research, as acute peripheral arterial ischemia is a pathology often underdiagnosed and undertreated adequately with potential fatal cardiovascular consequences.
I agree that the small size of population studied it is a significant limitation of the study, but I encourage the authors to extend the group of this category of patients, may be in a collaborative study and to consider this research as a pilot study.
Thank you for your thorough review and valuable comments on our manuscript. We appreciate your positive feedback and acknowledge the limitations of our study, including the small sample size. Your encouragement to extend the study to a larger population is greatly appreciated, and we will consider this suggestion for future research.
We are glad that you found the study to be original, scientifically interesting, and relevant to clinicians, and we appreciate your recognition of the importance of the topic. Your insight on the methodology, results, and discussion section is very helpful, and we will take your feedback into account as we work to further improve our work.
The authors have participated in the PERCEIVEStudy and are looking forward to further work on this area
Once again, thank you for your time and dedication to the peer-review process. We look forward to implementing your suggestions to enhance the quality of our research.

Round 2
Reviewer 1 Report
The authors have answered all of my questions and the paper has been greatly improved.
Reviewer 2 Report
no other comments